# Improving the In Vitro Removal of Indoxyl Sulfate and p-Cresyl Sulfate by Coating Diatomaceous Earth (DE) and Poly-vinyl-pyrrolidone-co-styrene (PVP-co-S) with Polydopamine

**DOI:** 10.3390/toxins14120864

**Published:** 2022-12-08

**Authors:** Stefania Roberta Cicco, Maria Michela Giangregorio, Maria Teresa Rocchetti, Ighli di Bari, Claudio Mastropaolo, Rossella Labarile, Roberta Ragni, Loreto Gesualdo, Gianluca Maria Farinola, Danilo Vona

**Affiliations:** 1Consiglio Nazionale delle Ricerche, CNR-ICCOM, Via E. Orabona 4, 70126 Bari, Italy; 2Institute of Nanotechnology, CNR-NANOTEC, Via Orabona 4, 70126 Bari, Italy; 3Department of Clinical and Experimental Medicine, University of Foggia, Via Pinto 1, 71122 Foggia, Italy; 4Dipartimento di Emergenza e Trapianto di Organi, Nefrologia, Dialisi e Unità Trapianti, Università Degli Studi di Bari “Aldo Moro”, Piazza G. Cesare 11, 70124 Bari, Italy; 5Visionage SRL, Via Marconi 19, 70011 Alberobello, Italy; 6Dipartimento di Chimica, Università Degli Studi di Bari “Aldo Moro”, Via E. Orabona 4, 70126 Bari, Italy; 7Consiglio Nazionale delle Ricerche, IPCF-CNR, Via E. Orabona 4, 70126 Bari, Italy

**Keywords:** diatomaceous earth, polyvinylpyrrolidone-co-styrene, chemical functionalization, polydopamine, uremic toxins, bioremoval

## Abstract

Polydopamine (PDA) is a synthetic eumelanin polymer mimicking the biopolymer secreted by mussels to attach to surfaces with a high binding strength. It exhibits unique adhesive properties and has recently attracted considerable interest as a multifunctional thin film coating. In this study, we demonstrate that a PDA coating on silica- and polymer-based materials improves the entrapment and retention of uremic toxins produced in specific diseases. The low-cost natural nanotextured fossil diatomaceous earth (DE), an abundant source of mesoporous silica, and polyvinylpyrrolidone-co-Styrene (PVP-co-S), a commercial absorbent comprising polymeric particles, were easily coated with a PDA layer by oxidative polymerization of dopamine at mild basic aqueous conditions. An in-depth chemical-physical investigation of both the resulting PDA-coated materials was performed by SEM, AFM, UV-visible, Raman spectroscopy and spectroscopic ellipsometry. Finally, the obtained hybrid systems were successfully tested for the removal of two uremic toxins (indoxyl sulfate and p-cresyl sulfate) directly from patients’ sera.

## 1. Introduction

Uremic toxins such as indoxyl sulfate (IS) and p-cresyl sulfate (PCS) (Figure 1) are gut microbiota-derived metabolites produced during metabolic processes, exclusively generated by degradation of dietary protein aminoacids, which accumulate in kidney compartments in patients with end-stage renal disease [1]. These toxic substances contain aromatic rings derived from tryptophan and tyrosine moieties, which are absorbed in the gut and sulfated via a sulfation process in the enterocytes and liver cells. PCS and IS circulate in the bloodstream tightly bound to albumin, and they are actively excreted in urine [2]. In chronic kidney disease patients, the blood concentration of IS and PCS increases as the renal failure worsens, so these metabolites can be used as predictors of poor clinical outcomes [3]. In addition, they are recognized as non-traditional cardiovascular (CV) risk factors [4]. Because of their albumin-binding properties, IS and PCS are not sufficiently removed by conventional dialysis methods, accumulating in ESRD patients. This is why material scientists are still developing different methods and materials for adsorbing IS and PCS in blood-fractioned serum. Strategies to remove these toxins are based on prolonged dialysis treatment [5], taking advantage of sophisticated synthetic high-flux membranes [6]. Different materials have been tested for removing these toxins, including poly-methacrylate (PMMA) [7], poly-ethersulfone/polyammide [8], and divinylbenzene [9]. Different coated and functionalized materials, useful in removing aromatic species similar to IS/PCS, have been reported in the literature: divinylbenzene coated with polyvinylpyrrolidone (DVB-PVP) [10], cellulose-bearing hexadecyl chains [11] and vitamin E-bonded cellulose [12] have been investigated for their hydrophobic nature and biocompatibility. However, the production of these composites is time- and energy-consuming, due to the use of co-precipitation processes, thermal extrusions or chemical copolymerization using eco-incompatible and bio-unfriendly conditions, together with the solvents’ toxicity. Here, we report the effect of a polydopamine (PDA)coating on the intrinsic capacity of the polyvinyl-pyrrolidone-co-styrene (PVP-co-S) co-polymer particles and diatomaceous earth (DE) to trap blood toxins. In particular, the efficiency of surface chemical modification of PVP-co-S and DE with polydopamine (PDA) was investigated in the adsorption, directly from blood sera, of IS and PCS as prototypes of protein-bound uremic toxins, which are currently the most studied in the context of the gut–heart–kidney axis.

## 2. Results and Discussion

Diatom microalgae and their fossil form (diatomaceous earth, DE) represent a low-cost and easily available source of nanostructured biosilica produced by single-celled microorganisms (diatoms) that uptake silicic acid from seawater via a complex metabolic biomineralization process to afford intricately patterned and hierarchically organized silica layers. In the last decades, biosilica from diatoms have been extensively studied for the possibility of functionalizing them, starting both from in vivo and in vitro methods [13,14,15,16] with a plethora of multifunctional molecules for applications ranging from biomedicine [17,18,19,20,21] to photonics and imaging [22,23,24].

Very recently, the in vitro removal of a series of phenolic compounds used as model pollutants was achieved by a surface chemical modification of diatomaceous earth with a polydopamine (PDA) layer easily produced in aqueous solution [25]. In fact, PDA exhibits excellent adhesive and entrapping properties [26,27] for different application fields.

In this frame, we aimed to investigate whether a PDA layer could improve the uremic toxin entrapping capacity of two adsorbent materials, such as DE and polyvynyl-pyrrolidone-co-styrene (PVP-co-S) co-polymer, a commercially available, nontoxic, totally organic bulk filler already proposed in the literature for dialysis processing [9].

The PDA coating of DE and PVP-co-S was easily obtained by in situ dopamine monomer oxidative polymerization in aqueous solution, in green conditions, without adding any toxic solvents or requiring further processing. In order to find the best entrapping properties, the formation of a single PDA coating layer around the support materials was compared with a double PDA coating by a full characterization with spectroscopic techniques in solution and solid state, and after use in the removal of indoxyl sulfate (IS) and p-cresyl sulfate (PCS).

### 2.1. In-Solution Characterization of PDA-Coated Adsorbent Materials: UV-Vis Spectra

The formation of single (+) and double (++) PDA coating layers both on DE and PVP-co-S was first assessed in solution by UV-Vis spectroscopy. After the PDA functionalization, a broad absorption peak appeared over 285 nm both in DE/PDA and PVP-co-S/PDA spectra, likely ascribable to the presence of the polymer on both supports (Figure 2). This broadened peak is typical of polyaromatic groups that are produced during the spontaneous oxidative polymerization of dopamine monomers in solution, and a general increased trend (in absorbance profiles) could be observed passing from one single layer to a double layer. PDA partially covered the intrinsic UV-Vis bands of PVP-co-S, whose values were set around 295, 390, 505 and 689 nm.

### 2.2. Solid State Characterization of PDA-Coated Adsorbent Materials: Raman Spectroscopy, AFM and Spectroscopic Ellipsometry

For a solid state characterization of the materials, all six solutions were spotted from hydroalcoholic solutions and dried on the specific support and subjected to Raman, SEM, AFM, and ellipsometric analyses.

Figure 3 shows Raman spectra of (a) bare DE and DE/PDA (single (+) PDA layer and double (++) PDA layer) and (b) pure PVP-co-S, PVP-co-S/PDA (single (+) PDA layer and double (++) PDA layer) recorded in the range 300–2000 cm^−1^.

The Raman spectrum of DE showed the Si-O modes, while the spectrum of PVP-co-S showed vibrational bands typical of C-O, C-C, C-N and C-H bonds.

Raman spectra of both DE/PDA (+ and ++) and PVP-co-S/PDA (+ and ++) showed a strong fluorescence masking most of vibrational signals and some weak CH, CC, CO and CN signals. No significant difference was highlighted by the Raman technique between the supports with one or two PDA layers.

All six solutions were spotted on a silicon surface and characterized by scanning electron microscopy (SEM) and atomic force microscopy (AFM).

Bare DE and DE/PDA (single (+) and double (++) PDA layer) were firstly investigated, and the resulting SEM images are shown in Figure 4.The bare DE sample was characterized by the co-presence of silica debris derived from pinnate and centrate diatom shells, commonly present in fossil biosilica. In the case of DE/PDA+ and DE/PDA++, the biosilica nanostructure appeared more compact and smoother as an effect of the PDA coverage.

The different superficial organization and coverage of PDA on DE were highlighted by atomic force microscopy through a combination of topographical and phase measurements.

Figure 4b shows 30 μm × 30 μm topographical and phase images of DE, DE/PDA+, and DE/PDA++. Topography of DE films showed a very rough surface, with a roughness that decreased from 421 nm to 282 nm with the increasing number of PDA layers, consistently with a decrease in DE porosity, as already revealed by SEM.

Contrarily, the phase image of DE was uniform, while both lighter and dark areas were present in the images of DE/PDA+ and DE/PDA++, indicating a chemical contrast that is ascribed to nonhomogeneous and sparse coverage by PDA.

Increasing the number of PDA layers, in the same scanned area, a higher number of dark areas were present, ascribable to the greater presence of polymer aggregates.

A zoomed 5 μm × 5 μm phase image of the DE/PDA++ highlighted how the PDA was distributed on even smaller areas.

Morphological characterization by AFM and SEM of DE samples showed that, even though the PDA did not seem to negatively affect the integrity of the structures both on large and smaller areas, increasing the number of PDA layers decreased the DE porosity, with a consequent increase in its clogging state, suggesting a lower capacity for entrapment of chemicals or pollutants. Moreover, PDA formed aggregates on DE. Both effects negatively affected the DE, whose required property as a material for pollutant removal is a marked nano- and microporosity conferred by a large surface area and free nanopores.

In Figure 5, SEM images of pure PVP-co-S and PVP-co-S/PDA (single (+) and double (++) PDA layer) are shown: the pure PVP-co-S appeared as a uniform, fused, smooth material on the silicon surface. Passing from 1 to 2 layers of PDA, nano/microfractures and nano/micropores appeared on the material surface.

In order to investigate the PDA organization on PVP-co-S, smaller areas, 60 μm × 60 μm, were measured by AFM, as shown in Figure 5. PVP-co-S morphology had a roughness that strongly decreased from 62 nm to 3.6 nm when the number of PDA layers was increased. Specifically, one layer of PDA yielded PDA aggregation phenomena on the PVP-co-S surface (PVP-co-S/PDA+), while uniform coverage was achieved by doubling the PDA quantity (PVP-co-S/PDA++).

The zoomed 5 μm × 5 μm topographic image of the PVP-co-S/PDA++ highlighted that PDA forms nanoparticles with a size of ~80 nm that covered the PVP-co-S surface uniformly. Moreover, phase images of the PVP-co-S/PDA++ were homogeneous, indicating a homogeneous PDA surface coverage.

Combining AFM and SEM measurements, we found that one PDA layer did not significantly affect the integrity of the PVP-co-S, and PDA did not cover it uniformly. The addition of a second PDA layer significantly affected the integrity of the PVP-co-S, increasing its porosity and, moreover, yielding uniform PVP-co-S coverage.

The modifications induced by the second layer of PDA, together with the hydrophobic nature of PVP-co-S, adjuvated by its poly-catechol and poly-hydroxy-indole moieties, suggest, contrary to the case of DE, an improved capacity for entrapment of chemicals or pollutants.

In order to derive the optical properties (refractive index and extinction coefficient) of pure PVP-co-S film and its thickness, and to quantitatively evaluate the PDA incorporation (%volume fraction) along the film thickness, PVP-co-S films with and without PDA deposited on glass were characterized by spectroscopic ellipsometry.

DE samples, due to their high roughness, were not eligible for reflection measurements and were not characterized by spectroscopic ellipsometry.

Figure 6a shows experimental ellipsometric spectra of the pseudo-refractive index, n, and extinction coefficient, k, of pure PVP-co-S, PVP-co-S/PDA+ and PVP-co-S/PDA++ deposited on glass. For PVP-co-S/PDA+, a broad peak at ~310 nm appeared and became stronger after doubling the number of PDA layers, a clear indication of PDA incorporation into the films (in agreement with absorption spectra in Figure 2b).

In order to estimate the optical properties of the pure PVP-co-S film from the ellipsometric measurements, a simple substrate/PVP-co-S film/air model where the optical properties of pure PVP-co-S film were derived through a Tauc–Lorentz dispersion equation with a layer thickness of 121 nm ± 10 nm, which was consistent with the maximum peak-to-valley distance, Rp-v, within the 60 μm × 60 μm scanned area (shown in Figure 4b) from AFM.

Figure 6b shows the spectra of the refractive index, n, and extinction coefficient, k, of the PVP-co-S layer derived by the ellipsometric analysis. Four main absorption bands, at 330 nm, 406 nm, 564 nm and 710 nm, were found, all shifted to larger wavelengths compared to absorption bands in Figure 2b (for example 330 nm instead of 320 nm, 406 nm instead of 400 nm, 560 nm instead of 540 nm and 710 instead of 700 nm) due to intermolecular electronic coupling in the solid state compared to suspension. n and k spectra of PVP-co-S were plotted in the same range of the absorption measurements shown in Figure 2b in order to make the comparison more immediate. The band at 330 nm is a characteristic spectroscopic feature of vinylderivate [29], the intense band at 406 nm is related to the π–π* transitions of its conjugated polymer backbones [30], the peak at 560 nm corresponds to the n-π* transitions [31], while the band at 710 nm is due to oxidized species [29]. The refractive index of pure PVP-co-S is in the range 1–1.5. Specifically, the refractive index of pure PVP-co-S at 633 nm, that is 1.35, is consistent with data reported in previous reports [32] for styrene-based materials, therefore supporting the validity of the present approach.

In order to quantitatively evaluate the PDA incorporation (%volume fraction) along the thickness in PVP-co-S+ and PVP-co-S++ films, experimental spectra in Figure 6a were modeled by a two-layer model, substrate/film/air, where the film was not a homogeneous layer but graded in composition, and it was fitted by combining the optical properties of three components, PVP-co-S, PDA and voids (that were introduced to simulate the roughness of the films), at the bottom and top of the graded film [28].

Specifically, we found that the graded film at the bottom was 100% PVP-co-S but finished at the top with a surface enrichment in PDA, whose volume fraction along the film thickness increased from 18% to 67%, while the void% decreased consistently with the decrease in film roughness according to AFM.

The working assumption that the film layer is a linear gradient in the refractive index was used once the simple fitting model assuming a homogeneous layer failed.

### 2.3. Removal of Uremic Toxins

The investigation of the capacity of the bulk materials to trap uremic toxins from human blood sera was performed using the system shown in Figure 7. Serum samples (s) were pumped by using an in/out 2-channel peristaltic pump (p) with a 0.5–15 mL/min flux speed. A lab-made (5 cm × 15 cm) cartridge-like glassy cylinder, containing different amounts of sorbing materials, was positioned after the p element, and a collection (c) processing in vials was used for HPLC analyses, coupled with mass spectrometry as explained in the experimental section.

Sorbent adsorption capacity for IS and PCS was expressed as removal percentages (%RR) (Figure 8, Figure 9 and Figure 10), exploiting a quantification approach as reported in the experimental section. The investigated variables of the samples were: use of single and double polydopamine coatings, mass of the sorbing bulk materials, kinetics definition using different times of adsorption of the toxic molecules. Considering the same quantity of sorbent (20 mg), after 30 min at room temperature, the single PDA layer increased the sorbent capacity of both DE and PVP-co-S compared to that of bare controls, while the double-coated PDA layer inhibited the adsorption of both IS and PCS only in the DE case (Figure 8). This phenomenon could be ascribed to a general clogging of the silica structures, which had been previously predicted.

When we look at the amount of sorbent material, taking into account the best performing materials (DE+ and PVP+, with one layer of PDA), an increase in quantity seemed to affect the adsorption capacity of both materials for IS and PCS, maybe due to a more pronounced packing effect of the matrix, which reduced the analyte diffusion into and out of the structures (Figure 9). Finally, the time of incubation mostly influenced the removal of both uremic toxins by PDA-DE rather than PVP-co-S (both with one coating of PDA, Figure 10). Both sorbents seemed to exercise their maximum function in the first 30 min, after which a gradual release of toxins was observed, more evident for DE.

## 3. Conclusions

The enhanced capacity of two different sorbent materials to entrap kidney secondary metabolites as a result of polydopamine coating was investigated in this paper. Polydopamine (PDA) is a bioinspired polymer easily synthesized via oxidative polymerization and aggregation of the water-soluble dopamine (DA) monomer using weakly alkaline conditions. The resulting polymer is soft, adhesive, biocompatible, suitable for further chemical functionalization [33], and fully poly-aromatic. Recently, this latter property was demonstrated to be useful for entrapping aromatic pollutants from model solutions. The presence of quinone and hydroxyindole moieties in the polydopamine structure has been exploited to produce stable PDA surface coatings capable of adsorbing and stably bonding aromatic pollutants via fast and simple π–π stacking interactions [34,35]. The possibility of using cheap support materials (DE, PVP-co-S) as bulk surface precursors together with the self-producible PDA polymer under environmentally friendly conditions (room temperature, short time reactions, absence of organic solvents) was described in this paper. Polydopamine coating does not require particular pretreatments of the silica particles or bulk polymers for material processing. Moreover, it is useful as a cheap, easy-handling polymer material compared to other industrially processed extractive polymers (polydimethylsiloxane (PDMS), polyacrylate (PA), polyethyleneglycol (PEG), divinylbenzene (DVB) coatings) [36].These commercial extractive layers need long processing, such as sol–gel and purification methods, electrospun or micro-deposition methods, and the use of organic/inorganic precursors, organic solvents, catalysts, and toxic alkoxysilanes as sol–gel precursors [37,38]. In contrast, PDA processing does not require toxic solvents, precursors, or catalysts, and the entire polymerization protocol is performed in aqueous buffer at room temperature, with molecular oxygen as an abundant, cheap, and ecofriendly oxidant agent for polymerization. For instance, in parallel we considered cellulose as a standard dialyzer material, using a past literature reference cited by the article [9]. With our material, which does not need any further optimization, we achieved performance similar to that for tested commercial, processed, and extracted cellulose, set for this biopolymer at 15–25% for indoxyl sulfate, and at 15–31% for p-cresyl sulfate. Moreover, the design of new organic polymer-based materials can be considered as cheap and easy as the optimization of removal techniques based on displacement phenomena. This last approach consists of triggering uremic toxin removal by altering the ionic strength [39] or by adding organic molecules [40] that compete with these toxins to occupy the primary albumin binding site. For instance, protein binding with uremic toxins is reversible and governed by non-covalent, mainly electrostatic or van der Waals forces, so the change in salinity or the insertion of a molecular competitor causes proteins to free toxins, facilitating the dialytic subtraction. As a future perspective, our proposed method could be combined with the displacement methodology by producing functional tailored polymers able to remove uremic toxins by mechanical/electrostatic interactions and, at the same time, bearingspecific competitive molecules directly bound to the sorbent material. In summary, this study shows that both diatomaceous earth (DE) and polyvinylpyrrolidone-co-styrene (PVP-co-S), as sorbent support materials, can be, in principle, exploited for the removal of uremic toxins. In the case of chemical surface functionalization with the polydopamine (PDA) moiety, coated composites are promising for the remediation of biological fluids through removal of kidney toxins. PDA effectively increased the % removal of both tested uremic toxins, in comparison with the uncoated bulk materials. The removal effect can be ascribed to the combination of the presence of the silica pores and π–π stacking interactions due to the PDA polymer in the case of DE, and the production of roughness and different surface polarity in the case of PVP-co-S particles. Our innovative PDA-coated materials can be, in principle, proposed as adsorbing systems for a cheap and green dialysis cartridge. The cartridge could be produced with the lone porous polymers or polymer/diatomite microparticles able to be infused by biological fluids and simultaneously remove uremic toxins by specific chemical interactions, for a combination of pore capture and surface adsorption. The cartridge could be exploited with a timing of the process compatible with dialytic treatment (<24 h). After integration into the classical dialytic geometry, our materials in cartridges could be chosen also for clinical trial protocol, thanks to their ascertained hemocompatibility and technical features.

## 4. Materials and Methods

### 4.1. Materials and Equipment

Tris·HCl, dopamine·HCl 98%, diatomaceous earth (DE) and polyvinylpyrrolidone-co-styrene (PVP-co-S) particles were purchased from Sigma Aldrich St. Louis, MO, USA). PCS potassium salt was synthetized by Organic Chemistry Laboratory of Dipartimento di Farmacia-Scienze del Farmaco, Università di Bari “A. Moro”, Consorzio C.I.N.M.P.I.S. IS potassium salt, ammonium acetate, methanol, acetonitrile, and distilled water of high–performance liquid chromatography grade (Ultra CHROMASOLV) were purchased from Sigma (St. Louis, MO, USA). PCS ammonium salt was from ALSACHIM (Bioparc, Illkirch, France), and indoxyl-4,5,6,7-d4 sulfate potassium salt was from Toronto Research Chemicals (North York, Toronto, ON, Canada).

### 4.2. Material Preparation

DE was pretreated with an acid-oxidative process as reported in the literature. DE and PVP-co-S particles (500 mg) were washed three times with 1 mL of NaCl solution (1% *w/v*) and collected via centrifugation (3000× *g* rpm, 10′) at each washing cycle. After 1 h of sedimentation, 3 mL of the Tris·HCl solution were added (50 mM, pH 8.5). The dispersion was kept under magnetic stirring at 350× *g* rpm and room temperature for 10 min. Then, dopamine·HCl (10 mg for 1 layer and 30 mg for 2 layers) was added, and the reaction mixture was stirred for 5 h. The polymerization process could be visually followed by paying attention to the color change of the matrix, from gray to black. Further water addition quenched the reaction, and both PDA-DE and PDA-PVP-co-S samples were collected by sedimentation and washed three times with distilled water.

### 4.3. Material Characterization

Instruments used for material characterization included: Axiomat microscope, Zeiss (Jena, Germany), in reflection mode, a confocal transmission microscope Leica SP8 X, UV-Vis Shimadzu UV—2401 PC spectrophotometer (Milan, Italy) and Fourier Transform Infrared-Attenuated Total Reflectance (FTIR-ATR) Perkin Elmer Spectrum Two Spectrophotometer (Rodgau, Germany). Raman spectra were collected using a LabRAM HR (Horiba-Jobin Yvon, Montpellier, France) spectrometer with 532 nm excitation laser under ambient conditions. Low laser power (<1 mW) was used to avoid heat-induced modification or degradation of the sample due to focused laser light during spectrum acquisition. The collection time was longer than 200 sec. The excitation laser beam was focused through a 50× optical microscope (spot size 1 mm, and working distance 1 cm). The spectral resolution was 1 cm^−1^. For AFM characterization, topographic and phase images were simultaneously acquired by the AutoProbe CP (Thermo-Microscope, Sunnyvale, CA, USA) microscope using a gold-coated Si tip with a radius of curvature < 10 nm and a resonant frequency of 80 kHz.in non-contact mode. Topographical images were used to determine the surface morphology and roughness (rms, root-mean-square roughness), while phase images highlighted the different chemisorption/packing of PDA on DE and PVP surfaces.

Optical characterization of films of pure PVP-co-S alone and with PDA was performed by spectroscopic ellipsometry (SE). SE spectra of the pseudo complex refractive index 〈N〉 = (n + ik) (where n is the real refractive index, and k is the extinction coefficient) and pseudoextinction coefficient were measured in the 1.5–6.5 eV (826–190 nm) range with a resolution of 0.05 eV at an incidence angle of 70° using a phase modulated ellipsometer (UVISEL, Jobin Yvon, Montpellier, France). To derive the thickness of PVP-co-S and its spectral dispersion of the refractive index, n, and the extinction coefficient, k, from the measured SE spectra of the pure PVP-co-S on glass, a simple substrate/homogenous film/air model fit analysis was used. Four Tauc–Lorentz oscillators described the main optical transitions of the PVP-co-S layer. From the optical modeling, a PVP-co-S thickness of 121 ± 8 nm was derived that was consistent with the AFM measurements. Once the optical function of the PVP-co-S was determined, and the PDA optical properties were known, as shown in Ref. [30], the measured SE spectrum of PVP-co-S with PDA films on glass were fitted using Bruggeman effective medium approximation assuming a simple substrate/graded film/air model to derive the PDA incorporation (%volume fraction) along the film thickness. Voids were needed to estimate the density of the films.

### 4.4. In Vitro Adsorption Experiment

To compare the efficacy of single (+) and double (++) PDA coating layers around DE and PVP-co-S to adsorb uremic toxins, for t0 determination, and testing different masses, we used serum samples (s) pumped by an in/out 2-channel peristaltic pump (p) with a 0.5–15 mL/min flux speed and a lab-made (5 cm × 15 cm) cartridge-like glassy cylinder containing different amounts of sorbing materials, positioned as testing element. For the different timing experiments, we coated a 24-well plate with the same amount of each tested adsorbent material. After cooling, 500 μL of pooled HD sera at known concentration of IS and PCS were added to the sorbent layer. Pre-HD blood samples were processed for routine clinical analyses. An aliquot of blood was centrifuged at 3000× *g* for 10 min to obtain serum samples, which were then stored at −80 °C until use. To measure total pCS and IS, pre-HD serum samples were pooled, centrifuged for 10 min at 10,000× *g* and 25 °C by using 10-kDa cutoff Amicon filter devices (Millipore, Billerica, MA, USA) and processed for liquid chromatography–mass spectrometry (LC-MS/MS) analysis, as previously described [6,10]. The pooled HD sera were left in contact with the sorbent materials at room temperature. Adsorption was calculated from the IS and PCS concentration remaining in the supernatants collected after 0.5, 1.0, and 4.0 h. IS and PCS concentrationswere measured by LC-MS/MS, as described below. Briefly, supernatants were collected and immediately centrifuged at 20,000× *g*, for 1 h at room temperature, to remove any residual sediment before mass spectrometry analysis. Adsorption experiments were carried out in triplicate for both uremic toxins.

Mass balance (MB) and percentage of removal (%RR) were calculated for both single (+) and double (++) PDA coating layers around DE and PVP-co-S. MB and RR were calculated from the measured initial and final concentration (C) as MB = (Ct − C0) × Total Volume; %RR = (C0 − Ct)/C0 × 100, with C0 = initial concentration, Ct = concentration of the supernatant at time t.

### 4.5. LC-MS/MS for Quantification of PCS and IS

IS and PCS were assayed in the experimental solution, before the addition of the sorbent material (C0), and after 0.5, 1.0, and 4.0 h of incubation at room temperature. Pre- and post-incubation IS and PCS supernatant levels were determined at each time point (0.5, 1.0 and 4.0 h). Total levels of IS and PCS were assayed by multiple reaction monitoring (MRM) analysis, using a triple quadrupole mass spectrometer (API4000, AB SCIEX, Carlsbad, CA, USA) equipped with an electrospray ionization source and connected to an on-line high-performance liquid chromatography system (CBM-20A LC, Shimadzu, Kyoto, Japan), as we previously reported [9]. All samples were run in triplicate.

### 4.6. Statistical Validation

A Mann–Whitney U test was used to validate differences recorded among uncoated and coated materials for IS and PCS removal. Results were considered statistically significant with *p* < 0.05.

## Figures and Tables

**Figure 1 toxins-14-00864-f001:**
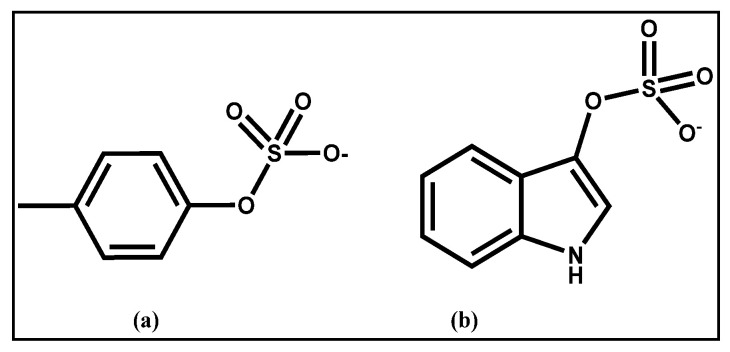
Model sulfated kidney toxins tested in experiments: (**a**) p-cresyl sulfate; (**b**) indoxyl sulfate.

**Figure 2 toxins-14-00864-f002:**
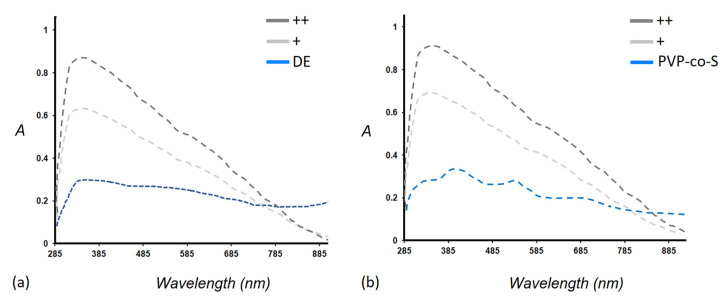
In-solution absorption spectra of (**a**) bare DE and DE/PDA (single (+) PDA layer and double (++) PDA layer); (**b**) pure PVP-co-S, PVP-co-S/PDA (single (+) PDA layer and double (++) PDA layer).

**Figure 3 toxins-14-00864-f003:**
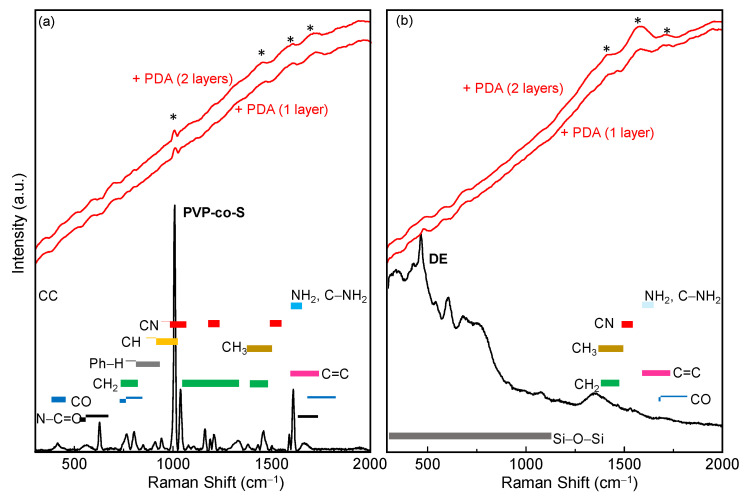
Raman spectra of (**a**) bare DE and DE/PDA (single (+) and double (++) PDA layer) and (**b**) pure PVP-co-S, PVP-co-S/PDA (single (1 layer) and double (2 layers) PDA). Colored bars assign Raman peaks to specific functional groups, while weak bands in PDA layers are indicated by (*).

**Figure 4 toxins-14-00864-f004:**
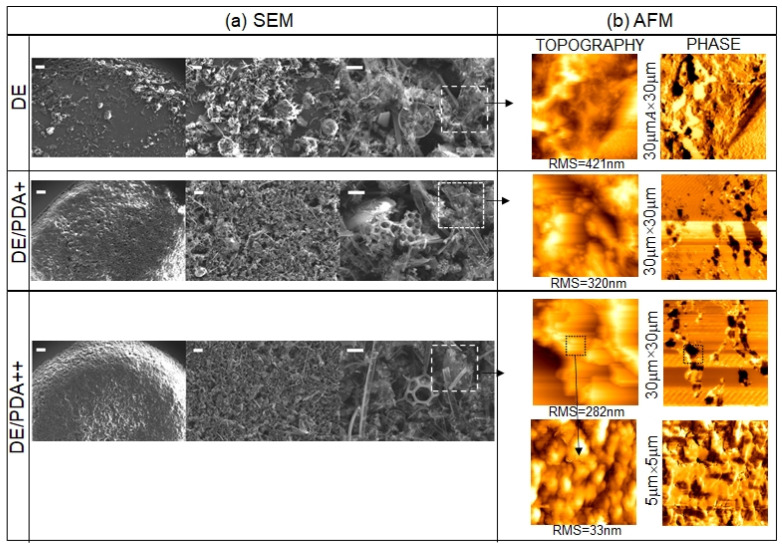
(**a**) Scanning electron microscopy images of bare DE, DE/PDA+ and DE/PDA++ films with different magnification (scale bars (left to right): 100, 20, 10 μm). (**b**) The corresponding 30 μm × 30 μm topographic and phase images. For DE/PDA++, zoomed 5 μm × 5 μm images are also shown. Roughness (RMS) values are also reported.

**Figure 5 toxins-14-00864-f005:**
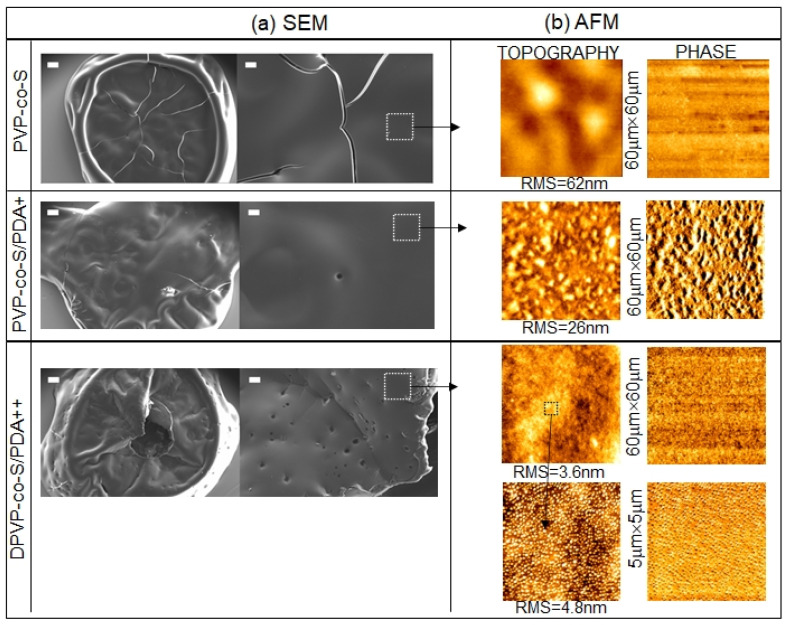
(**a**) Scanning electron microscopy images of pure PVP-co-S, PVP-co-S/PDA+ and PVP-co-S/PDA++ films with different magnification (scale bars (left to right): 100, 20 μm). (**b**) The corresponding 60 μm × 60 μm topographic and phase images. For PVP-co-S/PDA++, zoomed 5 μm × 5 μm images are also shown. Roughness (RMS) values are reported.

**Figure 6 toxins-14-00864-f006:**
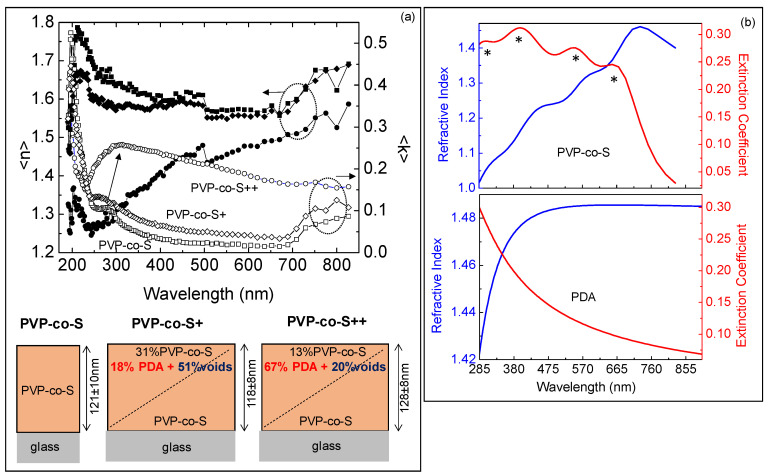
(**a**) Ellipsometric spectra of the pseudorefractive index, n (scattered filled points) and pseudoextinction coefficient, k (scattered empty points) of films of pure PVP-co-S (squares), PVP-co-S/PDA+ (rhombuses) and PVP-co-S/PDA++ (circles) deposited on glass. Structural models obtained by ellipsometric analysis at the bottom. In the ellipsometric models, the film thickness and the composition (% volume) of the bottom and top of the graded films in terms of volume fractions are shown. (**b**) Spectrum of the refractive index (blue) and extinction coefficient (red) of the PVP-co-S layer; the four main absorption bands are indicated by (*). The optical properties of PDA from [28] used in the modeling are also shown for comparison.

**Figure 7 toxins-14-00864-f007:**
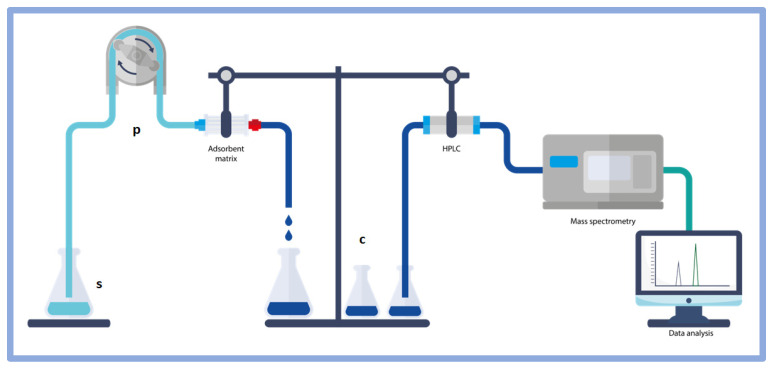
Scheme of the samples processing from the pumping to the HPLC investigation and detection.

**Figure 8 toxins-14-00864-f008:**
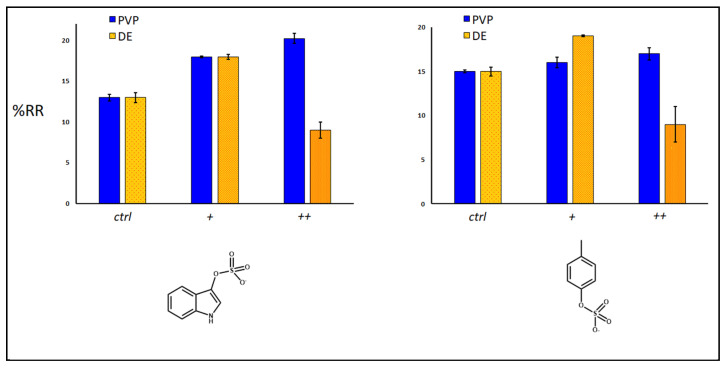
Percentage of removal (%RR) of IS and pCS in relation with coatings (+: 1 layer; ++: 2 layers). Control used: both uncoated materials (DE; PVP). Amount of each sorbent used: 20 mg; time of incubation at room temperature: 30 min. Mann–Whitney U test validation (left graph for PVP and DE): ctrl vs. (+) with *p* < 0.05; vs. (++) with *p* < 0.05; (right graph for DE): with *p* < 0.05. Significance for PVP only between ctrl and (++).

**Figure 9 toxins-14-00864-f009:**
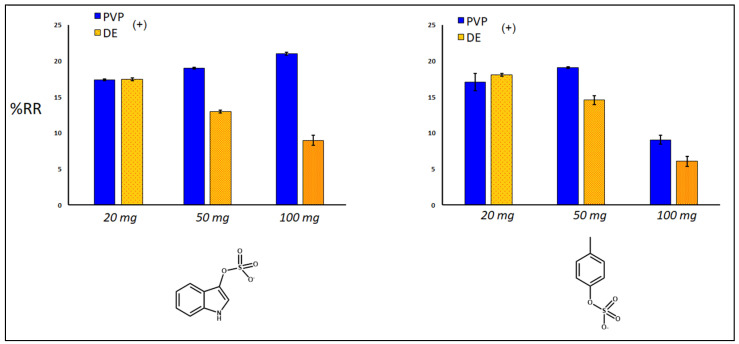
Percentage of removal (%RR) of IS and pCS in relation with different quantities of sorbent materials (20, 50, 100 mg). Time of incubation at room temperature: 30 min. Mann–Whitney U test validation (left graph for DE): (20) vs. (50) vs. (100) with *p* < 0.05; significance for PVP only between (20) and (100); (right graph for both PVP and DE): (20) vs. (50) vs. (100) with *p* < 0.05.

**Figure 10 toxins-14-00864-f010:**
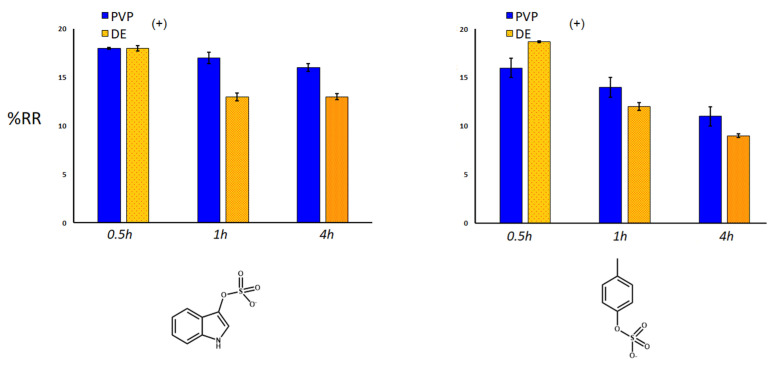
Kinetics of percentage of removal (%RR) for IS and pCS. Mass: 20 mg. Mann–Whitney U test validation (left graph for DE): (0.5) vs. (4) with *p* < 0.05; significance for PVP only between (0.5) and (4); (right graph for both PVP and DE): (0.5) vs. (1) vs. (4) with *p* < 0.05.

## Data Availability

Not applicable.

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
