# Peer review of "Improving the In Vitro Removal of Indoxyl Sulfate and p-Cresyl Sulfate by Coating Diatomaceous Earth (DE) and Poly-vinyl-pyrrolidone-co-styrene (PVP-co-S) with Polydopamine"

_toxins, 2022, doi:10.3390/toxins14120864_

Round 1

Reviewer 1 Report

In the present study the authors investigate the effect of pol- 45 ydopamine (PDA)coating on the intrinsic capacity of the polyvinyl-pyrrolidone-co-sty- 46 rene (PVP-co-S) co-polymer particles and diatomaceous earth (DE) to catch blood toxins, specially p-cresil sulfate and indoxyl sulfate.  This is aninteresting study that may shed some light on the removal of protein bound uremic toxins from the CKD patients bloodstream  Nevertheless, I have some concerns, on the please, find my comments below.

1. The authors describe that the used polymer is capable of remove IS and PCS, to this they performed a traditional characterization with UV-Vis, Raman, SEM and AFM. It is not possible to quantify shape retention (they need a better method such ass mass spectroscopy) and they have not tested the biocompatibility of PVP-cp-S. 

2. The better say that the binding polymer works, i suggest that the compare patients serum before and after dialyze with this polymer, as well as use some others circulatory bookmarks.

3. How the authors imagine to apply this method in the patients treatment?

4. line 347, please describe the IS and PCS concentrations used and why, justify.

Author Response

Letter to referee 1

Dear Editor,

we would like to thank the reviewer for his/her appreciation and critical comments. Her/his suggestions have contributed to improve our manuscript. We provide a revised version of the manuscript, following the reviewer’s indications. All changes in the MS are visible in track-change.

Below are the replies to the reviewer’s comment.

In the present study the authors investigate the effect of polydopamine (PDA)coating on the intrinsic capacity of the polyvinyl-pyrrolidone-co-styrene (PVP-co-S) co-polymer particles and diatomaceous earth (DE) to catch blood toxins, specially p-cresyl sulfate and indoxyl sulfate.  This is an interesting study that may shed some light on the removal of protein bound uremic toxins from the CKD patients bloodstream  Nevertheless, I have some concerns, on the please, find my comments below.

Question 1.1

The authors describe that the used polymer is capable of remove IS and PCS, to this they performed a traditional characterization with UV-Vis, Raman, SEM and AFM. It is not possible to quantify shape retention (they need a better method such ass mass spectroscopy)…

Answer 1.1

The point of investigating the presence of toxins directly in bound state on matrices after adsorption phenomena is important, and we thank the reviewer. Unfortunately, the greatest difficulty in carrying out this investigation lies in the nature of the support itself. In fact, it was not possible to exploit MALDI-TOF techniques due to the presence of the biosilica (DE), which is not conductive,  then a stripping of the polymer material from the bulk cannot be performed. Moreover, the polymer gives high intensity signals in MALDI compared to the small (not easy to strip) molecular toxins. Additionally, even mass spectrometry techniques directly on surfaces (e.g. Time of Flight Secondary Ion Mass Spectrometry (ToF-SIMS)) could not help due to i. the similar chemical π-backbone nature and composition of both the uremic toxins and the polydopamine (PDA) material and ii. the dilution effect of the adsorbed toxins over the poly-conjugated structures of the PDA matrix.

Our attempt to quantify toxins on matrix surfaces using the EDX-SEM investigation (using the sulfur atom as an atom probe) also failed, due to the very low concentration of toxins compared to the polymer.

For these reasons, we indirectly evaluated the removal efficacy of the two materials tested by LC-MS/MS analysis from the serum solutions before and after treatment.

Question 1.2

… and they have not tested the biocompatibility of PVP-co-S.

Answer 1.2 

Among materials exploited for fulfilling dialysis tubes or cartridge, our two discussed and investigated materials have been proposed ad hoc thanks to their high biocompatibility already reported in literature. For instance, PVP particles have already been introduced in the world of dialysis materials because of it decreased the inflammatory responses in patients treated with poly-sulfone-based cartridges [Problems in the evaluation of polyvinylpyrrolidone (PVP) elution from polysulfone membrane dialyzers sterilized by gamma-ray irradiation; doi: 10.1186/s41100-016-0047-x] Specifically, EFSA documents for security of some polymers as food additives, and as blends for medical treatments, including dialysis, state that “the Panel considered that toxicity studies available for PVP (and derivatives) showed no adverse effects at the highest doses tested”.  [Reevaluation of polyvinylpyrrolidone (E 1201) and polyvinylpolypyrrolidone (E 1202) as food additives and extension of use of polyvinylpyrrolidone (E 1201); doi: 10.2903/j.efsa.2020.6215].

Question 2

The better say that the binding polymer works, i suggest that the compare patients serum before and after dialyze with this polymer, as well as use some others circulatory biomarkers.

Answer 2

Thank you for this comment and for your suggestions. Indeed, in our work we compared total IS and PCS concentration in serum HD samples before dialysis, then after contact with the tested sorbent material as indicated by figures 8 and 9.  Due to the hydrophobic features of our coating materials (Diatomaceous Earth (DE)/PVV-co-S coated with a PDA layer- DE/PDA+) we tested their capacity to adsorb the albumin-binded uremic toxins, namely indoxyl-sulfate and p-cresyl-sulfate, which have an aromatic hydrophobic ring in their structure. We thought this could be enough for this kind of initial study. However, we agree that analysing the DE/PDA+ adsorption capacity of other circulatory biomarkers (such as albumin) can help to better characterize our sorbent material and this could be an interesting part of forthcoming studies, whose ultimate purpose will be to provide adsorbing systems for cheap and green dialysis cartridge.

Question 3

How the authors imagine to apply this method in the patients treatment?

Answer 3

The removal of uremic toxins is a crucial topic in the treatment of chronic nephropathic patients. These studies are useful for the technological implementation of dialysis cartridges which currently use polyvinylpyrrolidone for the removal of uremic toxins. In the study we demonstrated how to increase this capture. The self-assembling coating can be in principle and universally applied to all the bulk materials already commercially available.

For improving the quality of the work we added this sentence to the main text (pp.11-12, L364-372):

“Our innovative PDA-coated materials can be in principle proposed as fulfilling adsorbing systems for cheap and green dialysis cartridge. The cartridge could be produced with the lone porous polymers or polymer/diatomite microparticles able to be infused by biological fluids and simultaneously remove uremic toxins by specific chemical interactions, so a combination of pore capture and surface adsorption. The cartridge can be exploited with a timing of the process compatible with dialytic treatment (<24h). After the integration in the classical dialytic geometry, our materials in cartridges can be chosen in principle also for clinical trial protocol, thanks to their ascertained hemocompatibility and technical features.”

Question 4

…line 347, please describe the IS and PCS concentrations used and why, justify.

Answer 4

Samples analyzed consisted of patients sera, and the investigation did not occur on ad hoc produced model solutions. For this reason the %RR has been considered as a relative measure between before and after treatment, as reported in the experimental section.

Reviewer 2 Report

This is a experimental paper describing an innovative coating stategy for dialyser materials and analysing it´s removal potenzial for protein-bound (pb) solutes in dialysis process.

While being methodologically sound from a physician´s perspective (not being an expert chemist) the paper would gain more clinical interest, if the removal potenzial of the technique would be set in relation to existing or emerging strategies of pb solutes removal.  This is of particular importance, because such techniques could be combined to yield revolutionary pb solute removal currently not available.

On p1 l35 alternative removal strategies like ionic strength (PMID: 27259463) and molecular displacer strategies (PMID: 30755453) should be added for a more comprehensive picture.  In the discussion section, the results of the presented strategy may be discussed together with other, available strategies for removal of protein-bound solutes in meaningful combinations, to reach highest clinical efficacy.

In Fig. 8 and related text, controls should be described in more detail – was same material used, but uncoated ?  Was the use of standard diayser material considered as a control, and if no, why not ?

On p11, l370 “500 L of pooled sera at known concentrations …” – check 500 L, - for sure 500 mL is meant.  Characterization of blood sera must be given. Did they derive from animals or humans ? Was an ethical vote for use of human material obtained ?

Author Response

Letter to referee 2

Dear Editor,

we would like to thank the reviewer for his/her appreciation and critical comments. Her/his suggestions have contributed to improve our manuscript. We provide a revised version of the manuscript, following the reviewer’s indications. All changes in the MS are visible in track-change.

Below are the replies to the reviewer’s comment.

This is a experimental paper describing an innovative coating stategy for dialyser materials and analysing it´s removal potential for protein-bound (pb) solutes in dialysis process.

While being methodologically sound from a physician´s perspective (not being an expert chemist) the paper would gain more clinical interest, if the removal potential of the technique would be set in relation to existing or emerging strategies of pb solutes removal.  This is of particular importance, because such techniques could be combined to yield revolutionary pb solute removal currently not available.

Question 1

On p1 l35 alternative removal strategies like ionic strength (PMID: 27259463) and molecular displacer strategies (PMID: 30755453) should be added for a more comprehensive picture.  In the discussion section, the results of the presented strategy may be discussed together with other, available strategies for removal of protein-bound solutes in meaningful combinations, to reach highest clinical efficacy.

Answer 1 

We are thankful to the referee. We added the required references. Moreover, we added a parallel discussion about the potential and synergic possibility to combine our self-coating method with the displacement techniques (p.11; L348-355).

The direction of the discussion takes place as it follows: “Moreover, the design of new organic polymer-based materials can be considered as cheap and easy as the optimization of removal techniques based on the displacement phenomena.[40] This last approach consists of triggering the uremic toxins removal exploiting the infusion of small molecules which compete with these toxins to occupy the primary albumin binding site. As a future perspective, our proposed self-coating method can be of course combined with the incorporation or linkage of these competitive ligands directly on the dialysis materials through our polymer self-assembly.” Simply, our approach is a preliminary improvement, while the displacement technique relies a second step improvement, in combination with our proposal.

Question 2

In Fig. 8 and related text, controls should be described in more detail – was same material used, but uncoated ?  Was the use of standard diayser material considered as a control, and if no, why not ?

Answer 2

Agreeing with this referee, we added the control specification in the Figure 8 caption.

For our investigation we considered uncoated DE and PVP-co-S as controls, and in parallel we took account of cellulose as a standard dialyzer material,  using a past literature reference cited by the article [Efficacy of Divinylbenzenic Resin in Removing Indoxyl Sulfate and P-cresol Sulfate in Hemodialysis Patients: Results from an In Vitro Study and an In Vivo Pilot Trial (xuanro4-Nature 3.2); doi: 10.3390/toxins12030170]. In the cited article, removal by cellulose sets around 15-25% for Indoxyl sulfate, and at 15-31% for p-cresyl sulfate. These outcomes, already investigated in literature for this commercial material, are similar to those found for our bulk and (improved) coated materials.

However, in principle, contrary to extracted and processed cellulose, our proposed materials do not need long processing, they are not produced via specific organosol methods, such as sol-gel, but they can be produced via self-polymerization from aqueous media, and our purification methods consist of water washings and drying. Moreover, cellulose needs industrial texturing processings such as electrospun,  microdeposition methods or spin-fiberings.

We added a sentence (p11, L345-348) for discussing the comparison with this standard material.

Question 3

On p11, l370 “500 L of pooled sera at known concentrations …” – check 500 L, - for sure 500 mL is meant.  Characterization of blood sera must be given. Did they derive from animals or humans ? Was an ethical vote for use of human material obtained ?

Answer 3

We thank the review for his/her correction. In the revised version of the manuscript 500 L was corrected in “500 mL”. Serum samples derive from hemodialysis patients admitted to the Nephrology Unit of the Azienda Ospedaliero Universitaria Policlinico Consorziale di Bari which gave written informed consent for the use of this material for research as stated in the paragraph “Informed Consent Statement”, at the end of manuscript. As required, the following sentence, describing blood sera, has been added to the revised version of manuscript (p14, L444-451): “Pre-HD blood samples were processed for routine clinical analyses. An aliquot of blood was centrifuged at 3000´g for 10 minutes to obtain serum samples, which were then stored at -80 °C until use. To measure total pCS and IS, pre-HD serum sample were pooled, centrifuged for 10 min at 10,000g at 25 °C by using 10-kDa cutoff Amicon filter devices (Millipore, Billerica, MA, USA) and processed for liquid chromatography–mass spectrometry (LC-MS/MS) analysis as previously described [6,10] The pooled HD sera samples were left  in contact with the sorbent material at room temperature

Round 2

Reviewer 2 Report

Authors took my review concerns seriously and tried to implement new aspects.  However concerning reference [3], they missinterpreted this paper or least used a very missunderstandable wording to cite it "Their bioavailability strongly depends on ionic strength of plasma [3]."  In fact, not bioavailability but removal capacity increased by ionic strength.  And the ionic strength was experimentally added by infusion of high concentrated saline.  This is comparable to the method used in [40]  were ibuprofen instead of saline is the principle bringing toxins into the free fraction of sera and making them dialyzable.  I suggest to consider discussing such "displacer options" together and, again, to add reasoning if such free-setting methods of pb toxins or helpful in combination with dialyser layering described here.

I did not found ethical clearance of HD blood sera.  It would only be feasible to use these human materials under European data protection rights if these samples were completely anonymous. 

Author Response

Letter to referee 1

Dear Editor,

we would like to continue answering to our minor revision requests.

Question 1

Authors took my review concerns seriously and tried to implement new aspects.  However concerning reference [3], they missinterpreted this paper or least used a very missunderstandable wording to cite it "Their bioavailability strongly depends on ionic strength of plasma [3]."  In fact, not bioavailability but removal capacity increased by ionic strength.  And the ionic strength was experimentally added by infusion of high concentrated saline.  This is comparable to the method used in [40]  were ibuprofen instead of saline is the principle bringing toxins into the free fraction of sera and making them dialyzable.  I suggest to consider discussing such "displacer options" together and, again, to add reasoning if such free-setting methods of pb toxins or helpful in combination with dialyser layering described here.

Answer 1

As suggested, we made changes shifting the position of the literature references (from 3 to 39) and we discussed together the examples using a summarizing sentence:

P11, L333-344

“Moreover, the design of new organic polymer-based materials can be considered as cheap and easy as the optimization of removal techniques based on the displacement phenomena. This last approach consists of triggering the uremic toxins removal by altering the ionic strength [39], or by adding organic molecules [40] which compete with these toxins to occupy the primary albumin binding site. For instance, the protein binding with uremic toxins is reversible and governed by non-covalent, mainly electrostatic or van der Waals forces, so the change of salinity or the insertion of a molecular competitor causes proteins free toxins, facilitating the dialytic subtraction.  As a future perspective, our proposed method could be combined with the displacement methodology by the production of functional tailored polymers able to remove uremic toxins by mechanic/electrostatic interactions and, in the same time, bearing  specific competitive molecules directly bound to the sorbent material.”

Question 2

I did not found ethical clearance of HD blood sera.  It would only be feasible to use these human materials under European data protection rights if these samples were completely anonymous. 

Answer 2

Dear reviewer, the version of the paper you are supervising is cut in the last parts, but we ensure we already added this statement.

Please find here the sentence.

Regards

Informed Consent Statement: The present study was approved by the Independent Ethics Committee of the Azienda Ospedaliero Universitaria Policlinico Consorziale di Bari; all patients gave written informed consent for the use of this material for research purposes and the study was conducted in accordance with the Helsinki Declaration (Prot. N.4104/2013).
